# Comparative Value of CRP and FCP for Endoscopic and Histologic Remissions in Ulcerative Colitis

**DOI:** 10.3390/diagnostics14202283

**Published:** 2024-10-14

**Authors:** Oguz Kagan Bakkaloglu, Gozde Sen, Nuray Kepil, Tugce Eskazan, Enes Ali Kurt, Ugur Onal, Selcuk Candan, Melek Balamir, Ibrahim Hatemi, Yusuf Erzin, Aykut Ferhat Celik

**Affiliations:** 1Department of Gastroenterology, Kartal Kosuyolu High Specialization Education and Research Hospital, 34865 Istanbul, Turkey; 2Department of Internal Medicine, Cerrahpasa Medical Faculty, Istanbul University Cerrahpasa, 34098 Istanbul, Turkey; 3Department of Pathology, Cerrahpasa Medical Faculty, Istanbul University Cerrahpasa, 34098 Istanbul, Turkey; 4Department of Gastroenterology, Cerrahpasa Medical Faculty, Istanbul University Cerrahpasa, 34098 Istanbul, Turkey

**Keywords:** ulcerative colitis, CRP, histological remission

## Abstract

**Aim:** We have previously shown that CRP < 2.9 mg/L is a better predictor of endoscopic remission (ER) than CRP < 5 mg/L in ulcerative colitis (UC). Here, we prospectively evaluate CRP and FCP cut-offs and compare them in predicting ER and histological remission (HR) in UC. **Method:** One hundred thirty-five steroid-free UC patients were evaluated prospectively. ER was defined as Mayo endoscopic sub-score 0–1. In colonoscopy, the colon was evaluated as seven segments: rectum, sigmoid, descending, proximal-transverse, distal-transverse, ascending colon, and cecum. Two biopsies of each segment were evaluated for histological inflammation and graded using the Nancy and Geboes scores. All segment biopsies with Nancy < 1 and Geboes < 2 were defined as HR. **Results:** The optimum cut-off values for FCP and CRP were 120 μg/g and 2.75 mg/L for ER, respectively. AUC values of FCP and CRP were similar for ER and Mayo-0 disease in ROC analysis. CRP and FCP also had similar performances with these cut-offs regarding ER. While CRP was a predictor to assess the extensiveness of active UC, FCP was not. ROC analysis showed no difference between CRP and FCP regarding HR. Cut-off values for HR were 2.1 mg/L and 55 μg/g for CRP and FCP, respectively. CRP and FCP, in combination with the mentioned cut-off values, detected ER and HR in nearly 2/3 and ½ of the patients, respectively, with high specificity. **Conclusions:** Reappraised CRP (ER: 2.75 mg/L, HR: 2.1 mg/L) has as much diagnostic contribution as relevant FCP in predicting ER and HR and contributes more to revealing the proximal extension in active colitis compared to FCP. Relevant CRP and FCP combinations may improve the prediction rates.

## 1. Clinical Significance of the Paper?

When the appropriate cut-off value is used, CRP contributes as much as FCP in predicting both endoscopic and histologic remission.When used in combination, CRP and FCP can detect half of patients with histological remission non-invasively.CRP correlates with the proximal extension of UC and may be a better marker for evaluating the inflammatory burden.

## 2. Introduction

Endoscopic remission (ER) is accepted as a better predictor for long-term clinical remission and other ulcerative colitis (UC) outcomes than any clinical and biochemical parameters. However, in many cohorts, only half of all patients with clinical remission have ER at any time [1]. Especially in the biological era, this reality created a tendency among experts in inflammatory bowel disease (IBD), targeting ER to try to find predictors for ER without necessarily repeating colonoscopies. Furthermore, histologic remission (HR) is claimed to be the prognostic target for achieving even better long-term results than ER [2,3,4,5]. Colonoscopy is necessary for endoscopic and histologic assessment and is the obvious gold standard for mucosal evaluation. However, it may have many drawbacks, including inter-observer variability regarding endoscopic activity [6,7]. For this reason, the search for biomarkers or alternative methods for evaluating colonic mucosa continues, with popular attention mostly paid to fecal calprotectin (FCP). In this respect, considering the standard cut-off value, C-reactive protein (CRP) was ignored in terms of evaluating endoscopic remission due to its limited diagnostic contribution. However, there have been no attempts to reappraise and re-establish an optimized cut-off for CRP since low-level residual inflammation is often accepted as sufficient treatment success, especially for disciplines dealing with organ systems other than mucosal inflammation. UC is a type of inflammatory bowel disease in which the mucosal inflammatory intensity and extensiveness, except in rare submucosal involvement in severe cases and extraintestinal involvements, may both have some influence on the inflammatory burden. Therefore, mucosal healing or ER may be better represented by some particular lower CRP cut-offs than the standard one.

For the first time in the literature (UEGW 2019 Barcelona) [8], we have previously shown that a 2.9 mg/L cut-off for CRP was a better predictor of ER than the standard CRP cut-off (5 mg/L). Later, a detailed analysis, which was considered in a full paper, indicated that ER could be predicted when the CRP cut-off was further adjusted relative to the extensiveness of the mucosal involvement [9]. This later finding led us to consider that, despite the ER, ongoing histological activity might still influence CRP. Another retrospective study, albeit with a small number of cases, also confirmed our modified CRP cut-off [10].

In this prospectively designed study, we aimed to confirm the CRP cut-off value we had retrospectively defined before, compare the diagnostic performances of CRP (with a modified cut-off value) and FCP in predicting ER, and evaluate the contribution of CRP and FCP in predicting HR.

## 3. Methods and Materials

In this study, investigations and data collection were conducted prospectively with the consecutive enrollment of patients. One hundred fifty consecutive patients with UC, followed in the inflammatory bowel outpatient clinic, were enrolled in the study. Patients who lacked critical laboratory or endoscopic data were excluded, and a total of 135 patients’ data were analyzed cross-sectionally (Figure 1). The local ethics committee approved the study, which complied with the Declaration of Helsinki. The data underlying this article will be shared upon reasonable request to the corresponding author.

### 3.1. Study Group

Patients who were aged older than 18 years had a diagnosis of UC for at least 6 months, were followed up in the outpatient clinic, and volunteered to participate were included in the study. The treatments of the patients were evaluated. Patients using oral aminosalicylate (5-ASA) preparations, immunomodulatory drugs (azathioprine, methotrexate), anti-tumor necrosis factor (TNF) agents (infliximab, adalimumab, certolizumab), or other biologic agents (ustekinumab, vedolizumab) were included in the study. Patients currently using oral, intravenous, or rectal corticosteroids, or who had used steroids in the last 30 days for clinically active UC or any other reason, were not included in the study (corticosteroids can rapidly suppress inflammation and the acute-phase response, but mucosal healing may take longer.) Patients with any other possible cause for increased acute phase at the time of evaluation, such as chronic inflammatory conditions (arthritis, spondylitis) or infections (respiratory tract infection, urinary tract infection) other than UC, were also excluded from the study.

### 3.2. Clinical and Laboratory Assessment

Clinical and laboratory evaluations of the patients included in the study were performed 2–3 days before bowel cleansing for colonoscopy. The partial MAYO score [11], which includes stool frequency, rectal bleeding, and physicians’ global assessment, was used to define clinical activity. Clinical evaluations were performed by the same gastroenterologist. A partial MAYO score of <2 was accepted as clinical remission. The laboratory parameters evaluated were hemoglobulin (Hb), white blood cell count (WBC), neutrophil count, thrombocyte count, sedimentation rate, serum albumin, serum CRP levels, and FCP levels (30–1000 µg/g), which were measured using Quantum Blue^®^ fCAL (Bühlmann, Schönenbuch, Switzerland).

### 3.3. Endoscopic and Histologic Assessment

Colonoscopies were performed by two senior gastroenterologists involved in the study. The mucosal activity was graded as Mayo 0, 1, 2, and 3 based on endoscopic Mayo sub-scores [12]. Endoscopic remission was defined as an endoscopic Mayo score of 0 or 1. The Montreal classification was used to assess the extent of the disease [13]. Isolated ulcerative proctitis was grouped as E1, disease involving the distal colon up to the splenic flexure as E2, and extensions proximal to the splenic flexure were grouped as E3. Patients with normal mucosa were divided into groups based on the extent of their disease in their previous endoscopic evaluation.

The colon was evaluated as seven segments in colonoscopy as follows: cecum, ascending colon, proximal transverse colon, distal transverse colon, descending colon, sigmoid colon, and rectum. Two biopsies were planned from each segment for histologic evaluation. Biopsies were evaluated by a single pathologist who was blinded to the clinical, laboratory, and endoscopic findings. Validated Nancy and Geboes scores were used for histologic assessment, with Nancy < 1 and Geboes < 2 defined as HR [14,15]. Patients with all seven-segment biopsies compatible with remission histologically were considered the HR group. Patients with incomplete biopsies were evaluated in two ways. If active histology (Nancy ≥ 1 or Geboes ≥ 2) was present in the histologic evaluation, these patients were evaluated in the non-HR group. If biopsies were compatible with HR, these patients were excluded from the histologic assessment because any undisclosed histologic activity in the missing biopsies could not be excluded.

### 3.4. Statistical Analysis

The distribution characteristics of data were evaluated using the Kolmogorov–Smirnov test. CRP–FCP median and interquartile ranges were calculated for the groups; comparisons were made using independent samples tests, the Kruskall–Wallis, and the Mann–Whitney U test. The Spearman test was used for correlation analysis. CRP and FCP were evaluated using receiver operating characteristics (ROC) curve analysis in predicting ER and HR, and optimum cut-offs were determined. The sensitivity, specificity, positive predictive (PPV), and negative predictive values (NPV), as well as the diagnostic accuracy of these cut-off values, were calculated. The ROC area under the curve (AUC) values was compared using the Delong et al. method [16]. McNemar analysis was used to compare the diagnostic performance of the selected cut-off values, and Chi-square analysis was used to compare categorical data. The SPSS v. 29 software package (IBM Corp., Armonk, NY, USA) was used for statistical analysis. *p*-values of <0.05 were considered significant.

## 4. Results

One hundred thirty-five patients were evaluated in the study (Figure 1). The majority of the patients were male (60%), and the mean age was 46.2 (±13.9) years. Table 1 summarizes the study group’s demographic, disease, and treatment characteristics.

Approximately 70% of the patients were in clinical remission. ER (Mayo endoscopic sub-score 0–1) was present in 53.5% of the study group. There was no difference regarding age, disease duration, disease extension, and treatment in the ER group compared with endoscopic active disease (Table 1). ER was more common in the clinical remission group (*p* < 0.001). Hb levels were higher, while WBC, sedimentation rate, CRP, and FCP values were lower in patients with ER compared to patients with active disease; the difference was significant for Hb, CRP, and FCP (*p* < 0.001 for all).

ROC analyses were performed to evaluate the diagnostic power of CRP and FCP in predicting ER. The AUC for CRP was 0.784 [confidence interval (CI): 0.70–0.86], and the AUC for FCP was 0.849 (CI: 0.77–0.92) (Figure 2). Although the AUC of FCP was numerically superior, the AUC values of CRP and FCP did not differ significantly (*p* = 0.169).

In ROC analysis, the optimum cut-off value for predicting ER was calculated as 2.75 mg/L for CRP and 120 μg/g for FCP. The sensitivity and specificity of CRP for this cut-off value were 72% and 73%, respectively, and the sensitivity and specificity values for the FCP cut-off level of 120 μg/g were 79% and 77%. The diagnostic performances of CRP and FCP for these cut-off values are summarized in Table 2. The defined cut-off values of 2.75 mg/L for CRP and 120 μg/g for FCP were used, and their performances for predicting ER were similar (*p* = 0.74). In addition, in the clinical remission subgroup, these cut-off values of CRP and FCP had similar performances in predicting ER (*p* = 0.11). When combined, CRP and FCP had a 93% specificity (60% sensitivity) to predict ER, in consideration of the specified cut-offs.

Patients with active proctitis and active extensive colitis were compared regarding their CRP and FCP values. In the proctitis group, the median CRP was 4.6 mg/L (IQR: 4.1), and the median FCP was 333 μg/g (IQR: 892). In the extensive colitis group, the median CRP was 12 mg/L (IQR: 15.3), and the median FCP was 831 μg/g (IQR: 990). The difference between the CRP values of patients with proctitis and extensive colitis was significant (*p* = 0.04). Although the FCP was numerically higher in patients with extensive colitis, the difference was not significant compared with those with proctitis (*p* = 0.220) (Figure 3).

ROC analyses of CRP and FCP were also performed for predicting endoscopic Mayo-0 disease, as those calculated for ER (Mayo 0–1). The AUC for CRP was 0.718 (CI: 0.63–0.80), and the AUC for FCP was 0.816 (CI: 0.83–0.89). There was no statistically significant difference between the AUC values of CRP and FCP in terms of predicting Mayo-0 disease (*p* = 0.09) (Figure 1). The cut-off values calculated to predict Mayo-0 disease were lower than those for ER (2.3 mg/L for CRP and 80 μg/g for FCP). The diagnostic performance of these cut-off values in predicting Mayo-0 disease is summarized in Table 2.

Approximately half of the patients in ER (46%; n = 28) were also in HR. In some patients, active disease was noted in the proximal segments despite biopsies from the rectum being in HR. Histologic activity was noted in the sigmoid colon in 7 of 46 (15%) patients with HR in the rectum. There was histologic activity more proximally in 10 patients whose rectums and sigmoids were in HR. In total, although HR was detected in the rectum, 36% of these patients had active histology when the entire colon was considered. Regarding the segments in HR, endoscopically, the frequency of Mayo-0 and Mayo-1 was 93% and 7%, respectively. On the other hand, HR was detected in 69% of Mayo-0 segments, and HR was detected in 21% of Mayo-1 segments.

There was no difference in terms of age, disease duration, and extension (*p* = 0.55, *p* = 0.48, and *p* = 0.6, respectively) between the groups with and without HR. Among the treatment groups, HR was significantly more common in patients treated with biologics and azathioprine (41–20%, *p* = 0.02). CRP, FCP values, and total Nancy and Geboes scores of the colon segments of patients with and without HR are summarized in Table 3.

ROC analyses of CRP and FCP were performed regarding HR prediction; AUC values were calculated as 0.767 (CI: 0.66–0.87) for CRP and 0.797 (CI: 0.70–0.89) for FCP. The difference between CRP and FCP’s AUC values was not statistically significant (*p* = 0.62) (Figure 4).

The optimum cut-off values of CRP and FCP in predicting HR were calculated as 2.1 mg/L for CRP and 55 μg/g for FCP. These values were lower than those calculated to predict ER or Mayo-0 disease. Considering the CRP 2.1 mg/L cut-off value, the sensitivity, specificity, and diagnostic accuracy in predicting HR were 65%, 74%, and 71%, respectively, and 63%, 77%, and 74% for FCP. When CRP and FCP were combined, and both inflammatory markers were below the above-defined cut-off values, they showed a specificity of 95% and sensitivity of 50% in predicting HR.

Analyses were also performed to show the relationship between inflammatory load and CRP-FCP values. In these calculations, in which the colon was evaluated as seven segments, the sum of the endoscopic Mayo scores of the segments was moderately correlated with CRP and FCP (CRP, r: 0.501; FCP, r: 0.588; *p* < 0.001 for both). The sum of the Nancy and Geboes scores of the segments (histologic inflammatory load) were also moderately correlated with CRP and FCP (Nancy-CRP r: 0.503, Geboes-CRP r: 0.47; Nancy-FCP r: 0.566, Geboes-FCP r: 0.587; *p* < 0.001 for all). Nancy and Geboes scores were strongly correlated (r: 0.84, *p* < 0.001). Compared with Geboes, the Nancy score’s diagnostic accuracy in histologic remission was 92%.

## 5. Discussion

Our previous retrospective study showed that ER in UC could be predicted when an appropriate cut-off for CRP was chosen. This cut-off may also be adjusted further relative to the disease extension. In the present prospectively designed study, we demonstrated that FCP, which is very specific to intestinal inflammation, is not superior to CRP in predicting ER when the appropriate cut-off values were selected. It is also a novel contribution to the literature, revealing that approximately half of all patients with UC in HR could be detected with high specificity by combining CRP and FCP when the appropriate cut-offs were used.

Achieving endoscopic remission, rather than clinical remission, has been associated with decreased hospitalization, steroid need, and colectomy in UC [17,18,19,20]. Although colonoscopy is essential in evaluating mucosal status, patient preference, logistics, cost, and associated complications cannot be ignored. In particular, the search for alternative methods to evaluate symptom-free patients in clinical remission is at the forefront. Biomarkers such as CRP and FCP are frequently used for this purpose [21]. FCP is an anti-microbial protein found in the cytoplasm of neutrophils that can remain stable for some while in stool and is a sensitive way of checking the existence of leucocytes in the stool. FCP may be used as an assessment tool for any bowel inflammation both for diagnosis and follow-up [22,23]. More specifically, FCP is associated with the severity of endoscopic activity in inflammatory pathologies such as Crohn’s disease and UC and has been increasingly used in follow-up and treatment decisions [24,25,26]. However, FCP is far from being a gold-standard biomarker. The cost may prevent widespread use in some countries, and test results may show variability in different samples taken from the patient at the same time or day due to technical reasons [27,28]. The measurement method also has an impact on the result [29,30]. In addition, some patients may be reluctant to provide stool samples, despite not being so for blood samples.

We have shown in a retrospective series that CRP can predict ER better when a lower and appropriate cut-off value (2.9 mg/L) is used instead of the standard cut-off value (5 mg/L) (AUC: 0.854). It is stated that the most important hindrance of CRP is that it is not specific to intestinal inflammation. Although this is less of an issue for FCP, it can also be elevated in non-inflammatory bowel disease (IBD) pathologies [31]. Basal and inflammatory levels of CRP may be affected by genetics in some patients; population-wide genotype frequency is unknown [32]. This may put a small number of patients in the “poor responders” group regarding CRP. However, data on individual kinetics and genetic differences regarding FCP are scarce. Similarly, some variations can potentially affect FCP response. As we have shown retrospectively, CRP is strongly associated with endoscopic activity when factors that may be associated with extraintestinal inflammation are excluded. In the present study, we demonstrated the same correlation prospectively; considering the cut-off value of 2.75 mg/L, CRP was strongly associated with ER. Synchronously, the cut-off value for FCP in predicting ER was 120 μg/g. Different FCP cut-offs for ER in UC have been reported in the literature, ranging between 70–270 μg/g [33]. Different levels may be related to various reasons, such as the selected patient population, the definition of endoscopic remission, and measurement techniques, or they may reflect the large in-/inter-sample variance of FCP. In our study, although the diagnostic accuracy of FCP was numerically high, it was not found to be superior to CRP when the cut-off values, which we specified, were considered. This states that CRP (2.75 mg/L) is as guiding as FCP (120 μg/g) in predicting ER. Furthermore, the high specificity of CRP and FCP (at specified cut-offs), when combined together, can be a good alternative to endoscopy. Delta-CRP and Delta-FCP, which may be defined as the measure of reduction in inflammatory markers, would be important to overcome the inter-individual differences that apply to the CRP response, and they potentially exist for FCP. However, this study was not designed to show Delta-CRP and Delta-FCP values.

We previously reported that CRP, as a systemic inflammatory marker, was associated with the disease extension [9], and the cut-off for ER had the tendency to increase from rectal (E1) to extensive colitis (E3). In this prospective study, CRP levels were significantly higher in patients with active extensive colitis than those with active proctitis. However, this difference was nonsignificant for FCP. We also evaluated the total endoscopic and histologic inflammatory loads and their relationship with biomarkers. The decrease in cut-off values of CRP and FCP defined for ER (Mayo 0–1), Mayo-0 disease, and HR, in parallel with the depth of remission, indicates that these systemic and intestinal biomarkers are even correlated with subtle inflammation. The effect of the inflammatory load is also reflected in clinical outcomes at follow-up [5,23,34]. Regarding the inflammatory load–biomarker relationship, we showed a similar correlation for FCP as we did for CRP in our patient group, in line with the literature (r: 0.57) [35]. However, despite the correlation of FCP with an inflammatory load similar to CRP, we could not demonstrate a significant difference between FCP levels in patients with proximal extension when compared with active proctitis, as we mentioned above. Similar to our findings, in the study of Patel et al. evaluating FCP in UC, FCP was reported to be correlated with the inflammatory load. Still, the difference was nonsignificant regarding the Montreal disease-extension groups.

We agree with Stevens et al., who illustrated that in patients with UC, blood in stool, rather than diarrhea, is primarily associated with an increase in FCP levels [36]. This suggests that although the inflammatory load is associated with FCP, rectal inflammation may profoundly affect fecal biomarker levels, possibly due to proximity and residual blood and debris accumulation in the rectal area, even in patients with no rectal involvement. Relatively hard stool consistency in some patients with proctitis or distal colitis may increase rectal wall bleeding and debris rich in leucocytes. This creates a handicap for FCP, especially in UC. As a systemic inflammatory marker, CRP is not affected by these intraluminal issues, which can increase uncontrollably through the distal colon. Considering that UC affects the rectum in many patients, it will be difficult to interpret the proximal extension of UC by interpreting FCP values in active disease. However, based on the findings of our study, the increase in CRP parallel to the Montreal extension groups and its significance compared with FCP suggests that CRP has the potential to be used as a predicting marker of the proximal extension of UC, as well as disease activity.

HR, which is associated with better long-term results, emerges as a new treatment target [2,3,4,5]. It has been reported that histologic activity can continue despite ER and is associated with relapse [5,37]. This may be due to ongoing inflammation, as well as due to the misevaluation of ER. Therefore, the role of histologic evaluations will increase within the more precise definition of remission [4]. However, defining HR requires a colonoscopy [38] and is not devoid of inter- and intra-observer variabilities [39].

Another contribution of our study to the literature is the role of CRP in predicting HR and the comparison of its diagnostic performance with FCP. In terms of HR, the reported values for FCP range between 40–200 μg/g, covering the 55 μg/g cut-off we determined in our study [40]. The different reported cut-off values of FCP may be due to patient population characteristics, the types of kits used, and differences in the definition of HR. The fact that we have taken biopsies from all colon segments allows us to interpret the histologic activity in the colon and determine HR more precisely. Therefore, the FCP value, which we found close to the lower end of the range reported in the literature, may be more appropriate for histologic activity than the higher cut-offs reported by studies with a smaller number of cases or segmental biopsies. Rosenberg et al. reported that patients with HR might have a CRP value in the “normal range (<5 mg/dL)” [23]. The cut-off value that we found for CRP in terms of predicting HR was 2.1 mg/L. Similar to ER, we could not reveal a significant difference between the diagnostic performances of CRP and FCP in predicting HR. This indicates that CRP can be used for this purpose in clinical practice. Another point that should be emphasized is that HR can be predicted with very high specificity using both biomarkers, combined with the defined new cut-off values of our results. We think that this may have a place in daily clinical practice; using only a combination of CRP and FCP will practically predict HR in half of all patients.

In our study, although biopsies taken from the rectum indicated HR in some patients, active inflammation was observed in the biopsies of the proximal segments of the same patient. This suggests that there may be a patchy histologic activity. This may occur with the effect of treatment, or it may be related to variations in target segments and how biopsies are taken. As a result, taking a small number of biopsies may not fully reflect the histologic status. On the other hand, the fact that patients with Mayo-1 endoscopies can also be in HR is possible because a certain rate of endoscopist-related misjudgment is inevitable, and inter-observer variability may also pose endoscopic scoring problems [6]. The lower cut-offs calculated for Mayo 0, compared to Mayo 0–1, especially when considering the inter-observer variability in endoscopic scoring, suggest that Mayo 0 is a more appropriate indicator for assessing endoscopic healing than Mayo 0–1. It can be argued that neither endoscopic evaluation nor biopsy from a single segment to reflect the entire colon provides the opportunity to determine deep ER or HR with the uttermost accuracy. Taking routine biopsies from all segments to define HR comes with a higher workload and increased cost. On the other hand, inflammatory markers are devoid of endoscopists’ misjudgment and biopsy-related effects.

Endoscopic evaluation, with the help of artificial intelligence (AI)-assisted image recognition software, can potentially find a place in clinical practice. Many studies have been conducted on this subject, and AI appears to potentially increase diagnostic power while reducing observer-related variability [41,42]. However, there is still the need for colonoscopy and the additional cost associated with the software. Combining an easily accessible biomarker such as CRP with FCP can contribute to these systems.

The main limitations of our study can be listed as its single-center, cross-sectional design with a limited number of patients. However, the prospective data collection enabled us to eliminate the disadvantages of having a limited patient group. Another strength of the study is that histologic examinations were performed by the same pathologist, blind to clinical status and endoscopy by evaluating each segment of the colon, and the validated Nancy and Geboes scores were used in defining HR [43,44]. This approach provided us with the advantage of minimizing the impact of interobserver variability in HR assessments.

In conclusion, we have to adjust the CRP cut-off (2.75 mg/L) if we wish to use it more precisely to assess ER in UC. With these cut-off values, CRP and FCP have similar performances in predicting ER. CRP and FCP cut-off levels similarly correlate with colonic inflammatory load in patients with UC. However, CRP of active colitis seems to contribute more to revealing proximal extension compared with FCP. When appropriate cut-off values are selected in predicting ER and HR, there is no difference in diagnostic accuracy between CRP (ER: 2.75; HR: 2.1 mg/L) and FCP (120–55 μg/g). Combining an easily accessible biomarker such as CRP, which was non-inferior to FCP for the prediction of ER and HR in our study, can ease the clinical practice and allow us to limit costly and time-consuming procedures. Surprisingly, when CRP and FCP are combined in the above-mentioned limits, ER can be stated with very high specificity in 2/3 HR in half of all patients.

## Figures and Tables

**Figure 1 diagnostics-14-02283-f001:**
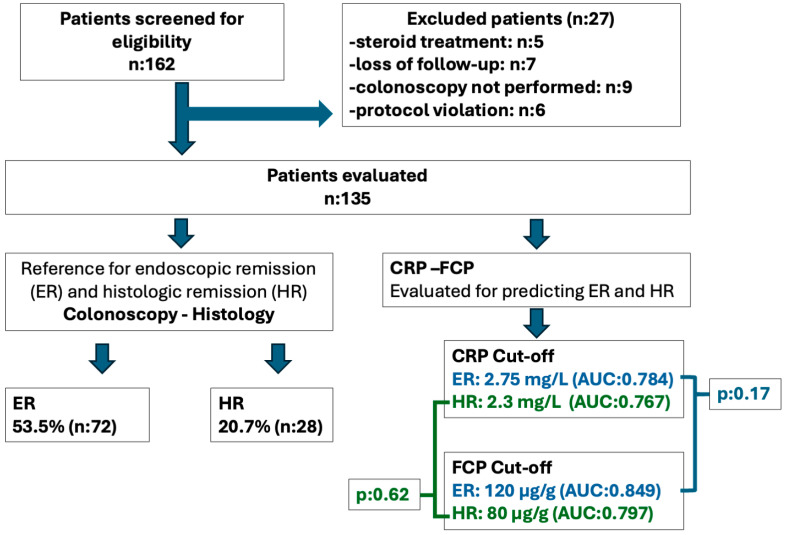
Participant flow diagram. One hundred sixty-two patients were screened and 135 were enrolled to study. Endoscopic remission (ER) and histologic remission (HR) were evaluated. CRP and FCP cut-off were determined, and their performances were compared in predicting ER and HR.

**Figure 2 diagnostics-14-02283-f002:**
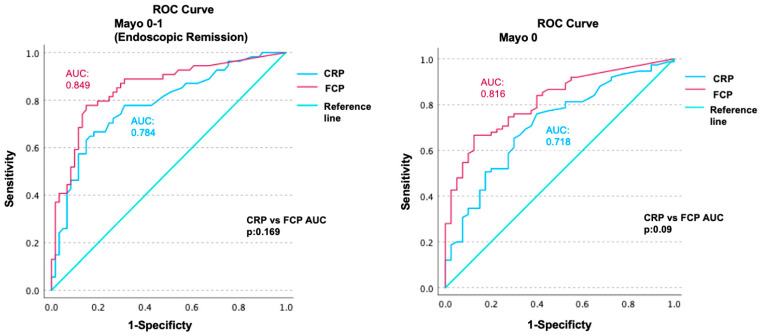
Receiver operating characteristics (ROC) curve analyses of CRP and FCP in predicting endoscopic remission (Mayo 0–1) and Mayo-0 disease. The AUC for CRP was 0.784 (confidence interval: 0.70–0.86), and the AUC for FCP was 0.849 (CI: 0.77–0.92) in endoscopic remission. The AUC for CRP was 0.718 (CI: 0.63–0.80), and the AUC for FCP was 0.816 (CI: 0.83–0.89) in Mayo-0 disease. There was no statistically significant difference between the AUC values of CRP and FCP for endoscopic remission (*p*: 0.169) and Mayo-0 disease (*p*: 0.09).

**Figure 3 diagnostics-14-02283-f003:**
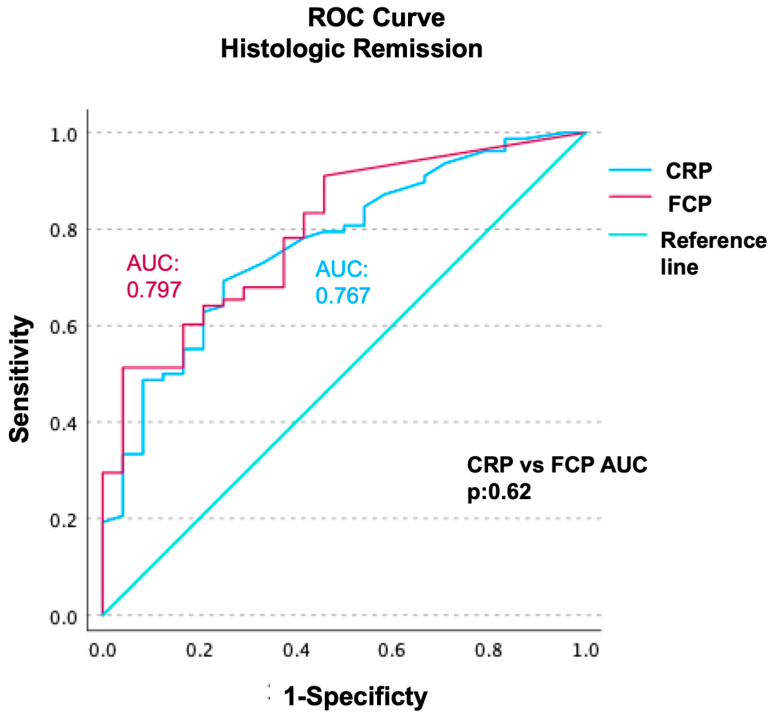
Receiver operating characteristics (ROC) curve analyses of CRP and FCP regarding HR prediction. AUC values were calculated as 0.767 (CI: 0.66–0.87) for CRP and 0.797 (CI: 0.70–0.89) for FCP. The difference between CRP and FCP’s AUC values was not significant (*p*: 0.62).

**Figure 4 diagnostics-14-02283-f004:**
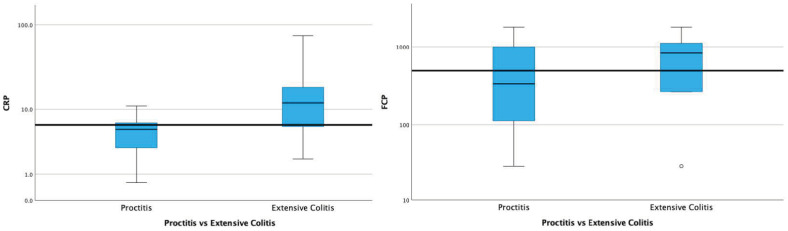
Comparison of patients with proctitis and extensive colitis in terms of CRP and FCP. In the proctitis group, the median CRP was 4.6 mg/L (IQR: 4.1), and the median FCP was 333 μg/mg (IQR: 892); in the extensive colitis group, the median CRP was 12 mg/L (IQR: 15.3), and the median FKP was 831 μg/mg (IQR: 990). The difference between the CRP values of proctitis and extensive colitis was significant (*p*: 0.04). The difference between the FCP values of proctitis and extensive colitis was insignificant (*p*: 0.220).

**Table 1 diagnostics-14-02283-t001:** The demographic, disease, and treatment characteristics of the study group and sub-groups regarding endoscopic activity.

		Whole Group (n:135)	Endoscopic Remission (ER) (Mayo 0–1) (n:72)	Endoscopic Active Disease (Mayo 2–3) (n:63)	*p*
Sex—M (%)		60	60	58.3	0.84
Age (mean ± SD)		46.2 ± 13.9	46.1 ± 13.1	46.6 ± 14.9	0.88
Age at diagnosis (median (IQR))		35 (20)	35 (19)	35 (22)	0.98
Disease duration		8 (11)	8 (11)	7.5 (10)	0.66
Disease extent—Montreal (%)					
	E1	12	13.9	8.3	0.6
	E2	48	47.2	50	
	E3	40	38.9	41.7	
Treatment (%)					
	Oral mesalazine	89.6	88.9	91.7	0.59
	Topical mesalazine	48.1	51.4	46.7	0.58
	Azathioprine (AZA)	40	41.7	40	0.84
	Biologic	30.5	31.9	30	0.81
	Biologic + AZA	20	22.2	18.3	0.58
	Anti-TNF	22.2	23.6	21.7	0.79
	Other biologics	8.3	8.3	8.3	1
Disease activity (%)					
	Clinical remission	69	87.5	51.7	<0.001
	Mayo 0–1 (ER)	53.3	100	0	-
	Mayo 0	35.6	66.7	0	-
Laboratory (mean ± SD, median (IQR))					
	Hb g/dL	13.5 ± 1.7	14 ± 1.6	12.9 ± 1.8	<0.001
	WBC/mm^3^	6800 (2500)	6300 (1700)	7300 (2770)	0.08
	SR mm/h	10 (14)	9 (13)	11.5 (16)	0.09
	CRP mg/L	2.7 (5.1)	2.1 (1.8)	5.4 (9.6)	<0.001
	FCP μg/g	125 (676)	51 (88)	613 (810)	<0.001

M: male; SD: standard deviation; IQR: inter-quartile range; Hb: hemoglobulin; WBC: white blood cell; SR: sedimentation rate; CRP: C-reactive protein; FCP: fecal calprotectin.

**Table 2 diagnostics-14-02283-t002:** The diagnostic performances and calculated cut-off values of CRP and FCP in predicting ER and Mayo 0 disease.

	Mayo 0–1 (ER)			Mayo-0		
Cut-Off Value	CRP 2.75 mg/L	FCP 120 μg/g	*p*	CRP 2.3 mg/L	FCP 80 μg/g	*p*
AUC	0.784	0.849	0.169	0.718	0.816	0.09
Sensitivity	72	79		70	75	
Specificity	74	77		66	70	
PPV	69	75		78	82	
NPV	77	80		56	59	
D.Acc	73	78		68	73	

ER: endoscopic remission; CRP: C-reactive protein; FCP: fecal calprotectin; AUC: area under curve; PPV: positive predictive value; NPV: negative predictive value; D.Acc: diagnostic accuracy.

**Table 3 diagnostics-14-02283-t003:** CRP, FCP values, and total Nancy and Geobes scores of colon segments of the patients regarding histological and endoscopic activity.

	Histological Active Disease		Histological Remission		
Median (IQR)	Endoscopic Active	Endoscopic Remission		*p* ^δ^	*p* ^ϕ^
CRP	5.4 (9.6)	2.3 (1.2)	1.6 (0.9)	0.043	0.002
FCP	613 (810)	59 (127)	29 (39)	0.024	<0.001
Total Geobes Score	14 (13)	8 (9)	0 (1.3)		
Total Nancy Score	7 (10)	4 (7)	0 (0)		

*p*^δ^: histological remission vs. endoscopic remission (histological active). *p*^ϕ^: histological remission vs. histological active (regardless of endoscopy). CRP: C-reactive protein; FCP: fecal calprotectin; IQR: inter-quartile range.

## Data Availability

The data underlying this article will be shared on reasonable request to the corresponding author.

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
