# Peer review of "Comparative Value of CRP and FCP for Endoscopic and Histologic Remissions in Ulcerative Colitis"

_diagnostics, 2024, doi:10.3390/diagnostics14202283_

Round 1
Reviewer 1 Report
Comments and Suggestions for Authors
The paper is interesting and may be a valuable contribution to the reserach and interpretaion of biomarkers in IBD. I have enclosed some comments and suggestions for improvement in the enclosed file. In general I would modify the conclusion somewhat and acknowledge the need for confirmatory studies to establish cut-offs. An elaboration on the importance of patients that do not produce CRP may be helpful for the general reader. I support the conclusion that a combination of CRP and FCP is useful in clinical practice, and in my opinion this emerges as the main finding of the study.

There are some inadequacies in the quality of English language - and editing is mandatory.
Reviewer 2 Report
Comments and Suggestions for Authors
The aim of the study is to evaluate the ability of the commonly used biomarkers CRP and fecal calprotectin in predicting endoscopical (ER) and histological (HR) remission in patients suffering from ulcerative colitis. This is done by including 135 patients suffering from ulcerative colitis which all had colonoscopy with assessment of ER according to the Mayo score and HR according to Geboes and Nancy scores. The conclusion is that low concentration CRP and calprotectin can predict both ER and HR with reasonable precision.
This conclusion is fair based on the evidence provided. The dogmatic view regarding the use of biomarkers have been that CRP has limited value as a marker in standard concentrations. For that reason it is an original thought to evaluate its usefulness in lower concentrations and to compare it with calprotectin. The results of the study are of relevance in the daily clinic and may contribute toa reduction in the numbers of invasive procedures like ileocolonoscopy performed.
However a few comments and suggestions are appropiate.
First of all the authors quote that this is a prospective study. Actually it is not. The patients are included consecutively but the study is cross sectional in the sense that the patients are not follewed over a period of time.
The authors define ER as a Mayo endoscopic subscore of 0-1. Whether Mayo 1 should be considered as remission can be questionned and the authors actually document in the paper that the rate of HR is much higher in Mayo 0 than in Mayo 1 which suggests that Mayo 1 is not really remission. It is a clear strength that separate analysis were performed for Mayo 0 and Mayo 0-1 but the discussion should include some remarks regarding the subject of assessing endoscopic healing. Besides central reading was not applied in the documentation of the scores. It is well known that interobserver variation exists in Mayo scoring. This issue should also be touched upon in the discussion
Regarding the assessment of the Nancy and Geboes scores were all samples evaluated by the same pathologist ? Also in the assessment of histological scores interobserver variation can occur.
It is mentioned in the results section that Hb was lower in patients with ER compared to patients with patients with active disease. This is wrong it is higher
In table 1 at the top a line should be included showing the actual number of patients in the different groups.
There is a disturbing wrong link between the figure legends and the figures regarding fig 3 and 4.
Reviewer 3 Report
Comments and Suggestions for Authors
The current paper compared the value of CRP and FCP in prediction of ER and HR for UC, which is significant in clinic use. We found the similar trend in clinic that FCP is more difficult to be decreased than the CRP after therapy, which means FCP should be more specific to predict UC's healing. However, there are several issues to be solved.
1.The conclusion should be presented as the most significant results, such as which is better to predict the HR or ER.
2.In the abstract, it is said Cut-off values for 28 HR were 2.1mg/L and 55μg/g for CRP and FCP respectively, but in the figure 1, there are differences.
3.Figure 4 should show the ROC curve, but there is a box plot.
4.Figure 2 showed FCP is superior to CRP in predicting ER, but this is not included in the conclusion.
5.What is the performance of combination of CRP and FCP, it may be better than by each of them.
